# Mapping trends in insecticide resistance phenotypes in African malaria vectors

Penelope A. Hancock[1]*, Chantal J. M. Hendriks[1], Julie-Anne Tangena[2], Harry Gibson[1], Janet Hemingway[2], Michael Coleman[2], Peter W. Gething[3,4], Ewan Cameron[1], Samir Bhatt[5], Catherine L. Moyes[1]*

1 Big Data Institute, University of Oxford, Oxford, United Kingdom, 2 Department of Vector Biology, Liverpool School of Tropical Medicine, Liverpool, United Kingdom, 3 Telethon Kids Institute, Perth Children's Hospital, Perth, Australia, 4 Curtin University, Bentley, Perth, Australia, 5 Department of Infectious Disease Epidemiology, Imperial College, St Mary's Hospital, London, United Kingdom

* hancock.penelope@gmail.com, (PAH); catherinemoyes@gmail.com (CLM)

## Abstract

Mitigating the threat of insecticide resistance in African malaria vector populations requires comprehensive information about where resistance occurs, to what degree, and how this has changed over time. Estimating these trends is complicated by the sparse, heterogeneous distribution of observations of resistance phenotypes in field populations. We use 6,423 observations of the prevalence of resistance to the most important vector control insecticides to inform a Bayesian geostatistical ensemble modelling approach, generating fine-scale predictive maps of resistance phenotypes in mosquitoes from the *Anopheles gambiae* complex across Africa. Our models are informed by a suite of 111 predictor variables describing potential drivers of selection for resistance. Our maps show alarming increases in the prevalence of resistance to pyrethroids and DDT across sub-Saharan Africa from 2005 to 2017, with mean mortality following insecticide exposure declining from almost 100% to less than 30% in some areas, as well as substantial spatial variation in resistance trends.

## Introduction

Insecticide resistance in African malaria vector populations has serious consequences for malaria prevention. Long-lasting insecticide-treated nets (LLINs) have achieved substantial reductions in malaria prevalence thus far in Africa [1], but the number of insecticides currently available for use in LLINs is very limited. Until recently, pyrethroids were the only class approved for use in LLINs, and recently launched new generation nets still use pyrethroids in combination with either an insect growth regulator, a pyrrole, or a synergist that inhibits the primary metabolic mechanism of pyrethroid resistance within mosquitoes [2,3]. A wider range of options is available for indoor residual spraying (IRS), but pyrethroids are less expensive than many alternatives and are still used for IRS in malaria-endemic sub-Saharan African countries [4,5].

**Data Availability Statement:** The predictive maps of the mean prevalence of resistance are available to download from Figshare (10.6084/m9.figshare. 9912623) and will be available to visualise on the

Malaria Atlas Project website (https://map.ox.ac.uk/explorer/#). The susceptibility test data is available to download (https://doi.org/10.1101/582510 [8]). Sets of susceptibility test data and predictor variable data in the form used by the statistical modelling analyses are available from GitHub. Numerical data corresponding to the manuscript figures are available to download from Figshare (10.6084/m9.figshare.9912623). R code for implementing the extreme gradient boosting, random forest, and boosted generalized additive models and the R-INLA geostatistical models is available on GitHub at 10.5281/zenodo.3751786 (Hancock, 2020).

**Funding:** This work was funded by Wellcome Trust Grant 108440/Z/15/Z (CLM). The funders had no role in study design, data collection and analysis, decision to publish, or preparation of the manuscript.

**Competing interests:** The authors have declared that no competing interests exist.

**Abbreviations:** BGAM, boosted generalized additive model; CI, credible interval; GMRF, Gaussian Markov random field; IHS, inverse hyperbolic sine; IRS, indoor residual spraying; ITN, insecticide-treated net; LLIN, long-lasting insecticide-treated net; MAE, mean absolute error; PET, potential evapotranspiration; PIT, probability integral transform; RF, random forest model; RMSE, root mean square error; Vgsc, Voltage-gated sodium channel; XGB, extreme gradient boosting model.

Although there is evidence that pyrethroid resistance in African malaria vector populations is increasing [6,7], the wide array of field studies that are available do not provide a spatially comprehensive time series of resistance trends [8]. Quantifying these trends will improve our understanding of the historical spread of resistance and assist in designing insecticide resistance management strategies [9]. Comprehensive spatiotemporal analyses of resistance are also necessary to facilitate its inclusion in epidemiological models of malaria that inform decision-making at national and global levels [9]. Efforts to estimate trends in insecticide resistance are impeded by limitations associated with the available observations of resistance phenotypes in field mosquito populations. Observations from standardized susceptibility tests, which indicate the prevalence of phenotypic resistance in field populations, cover a wide geographic area and span several decades [8,10]. However, the spatial coverage of these data is sparse and heterogeneous, and resistance has rarely been monitored consistently over time, meaning that very few time series are available [9]. Moreover, these susceptibility tests have a large measurement error, and replication is required to robustly estimate resistance phenotypes.

Several studies have reported spatial variation in resistance within a country, but these studies haven't analysed the spatial trends behind this variation [11,12]. In addition, time series of resistance data have shown changes over time at the location of sentinel sites [13–20]; however, only one study has investigated temporal trends across Africa [7]. This study generated a single trend over time for the whole continent, for each species and insecticide, and did not investigate differences in trends between locations. No previous study has analysed spatial variation across multiple countries or investigated spatiotemporal trends within these regions.

Our capacity to understand and predict insecticide resistance can benefit from considering the variables that may influence selection for resistance. Sources of insecticides in the environment include the application of insecticide-based vector control interventions for public health—such as LLINs and IRS—and the application of agricultural insecticides, which include the same insecticide classes as those used in vector control [21]. LLIN coverage increased markedly across Africa from 2005 in response the global Roll Back Malaria initiative [22, 23], while IRS coverage has been restricted to much smaller areas [4]. Early insecticide-treated nets (ITNs) used permethrin or deltamethrin, whereas α-cypermethrin is now the most commonly used pyrethroid in LLINs. Over the last 20 years, deltamethrin, λ-cyhalothrin, and DDT have been used for IRS, with α-cypermethrin first used in mass campaigns in 2003. Since 2015, deltamethrin has been the only pyrethroid reported to be used in mass IRS campaigns along with DDT and other non-pyrethroid insecticides [4]. Several studies have demonstrated a local increase in insecticide resistance in field mosquito populations following the implementation of LLINs, IRS, or both [19,20,24–26] although in other locations evidence of higher resistance after the introduction of these interventions was not found [24,27]. Associations between agricultural pesticide use and insecticide resistance have also been found [21,28], and there is evidence that pesticide contamination of water bodies is a source of selection pressure for resistance acting on mosquito larvae [29]. Relationships between resistance and drivers of selection will, however, vary geographically depending on population structure [30,31]. Genetic mechanisms of resistance also differ across mosquito species [25,30], and even closely related mosquito species have different ecological niches [32,33], as well as different blood feeding behaviour and preferences, meaning that they are likely to experience differences in insecticide exposure [34].

Pyrethroid resistance has not spread outwards from a single origin, and there are several known resistance mechanisms. The mechanisms that have been identified can be classed into target site mutations, up-regulation of detoxifying enzymes, and cuticular thickening [35–38]. Each class is associated with multiple genetic variants that confer resistance, and there is evidence that at least some of these have arisen independently multiple times across Africa [39].

Once a new variant has arisen, increases in its frequency are driven by selection pressures that differ from place to place, for example, through the spatially varying coverage of pyrethroid use in public health and agriculture [4,22,40]. The spread of these genetic variants to new locations is driven by gene flow that also varies with location [30,41–45]. Here, we consider a species complex, and the "spread" of resistance is further influenced by introgression of resistance genes from one species to another [46–51]. The geospatial patterns in the pyrethroid resistance phenotype that we are studying here are, therefore, the result of a complex combination of multiple origins of multiple genetic variants that have then spread through gene flow and introgression.

There is evidence that some mechanisms of pyrethroid resistance differ in the level of resistance they confer to different compounds; for example, different depletion rates were found across 6 pyrethroid compounds when they were metabolised by 8 *Anopheles* cytochrome P450 enzymes under controlled conditions [52]. These differences have not been investigated for all mechanisms of resistance, and there is no evidence that the differences seen for individual resistance mechanisms tested in the laboratory translate into diverging patterns of resistance in the field.

To develop predictive models of insecticide resistance in field populations that can represent variable, nonlinear interactions with environmental, biological, and genetic variables, we utilise an ensemble modelling approach. The approach exploits the multifaceted strengths of different modelling methodologies, using machine-learning methods to extract predictive power from a set of covariates and then allowing a Bayesian geostatistical Gaussian process to model the autocorrelated residual variation [53]. Bayesian geostatistical models provide a robust model of residual autocorrelation that can be applied to spatiotemporal data with a heterogeneous sampling distribution [54]. Their application to observations from insecticide susceptibility tests conducted over a range of locations across Africa has previously demonstrated broad-scale associations between resistance to different types of pyrethroids, as well as the organochlorine DDT [55]. The models developed in this study exploit these associations in resistance across different insecticides to improve resistance predictions for individual insecticide types.

Using a database containing the results of standard insecticide susceptibility tests performed on mosquito samples collected throughout Africa [8], we extracted the results of 6,423 tests conducted on samples from the *A. gambiae* species complex, which are among the most important African malaria vectors. We used this data set in our model ensemble to quantify variation in the prevalence of resistance to pyrethroids and DDT over the period 2005–2017 by developing a series of predictive maps. Our models are informed by a suite of potential explanatory variables describing the coverage of insecticide-based vector control interventions, agriculture and other types of land cover, climate, processes determining the environmental fate of pesticides, and the distribution of the sibling species that make up the *A. gambiae* complex. Our results show dramatic changes in insecticide resistance phenotypes in malaria vector populations across Africa over a 13-year period and identify variables that were important in shaping these predictions.

## Results

### Spatiotemporal trends in the prevalence of insecticide resistance

**Pyrethroid resistance.** We investigated spatiotemporal trends in the prevalence of phenotypic resistance in the *A. gambiae* complex to 4 pyrethroids: deltamethrin, permethrin, λ-cyhalothrin, and α-cypermethrin. Due to the lack of observations from central Africa, we partitioned the data into 2 separate spatial regions covering western and eastern parts of the

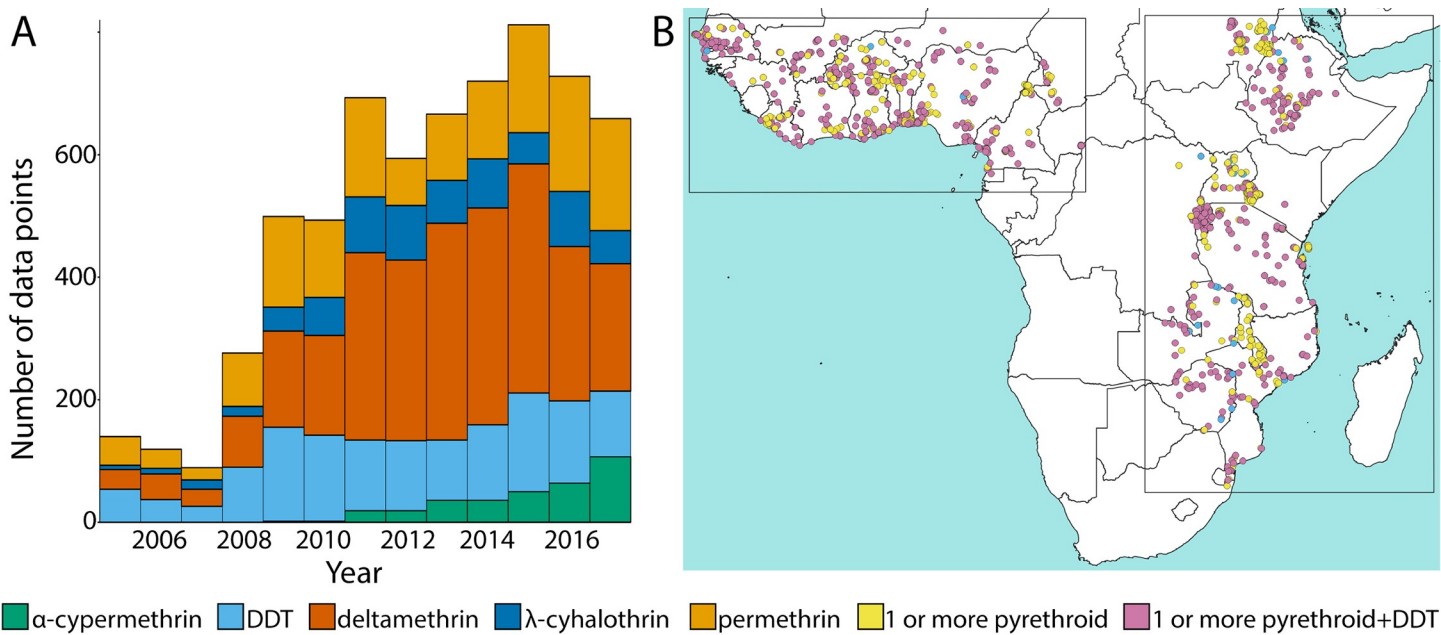

**Fig 1. The sampling distribution of the pyrethroid and DDT susceptibility test observations for mosquitoes from the _A. gambiae_ species complex across space and time.** (A) The number of susceptibility test observations for each year and insecticide type. Numerical values are provided in S1 Data (10.6084/m9.figshare.9912623). (B) The sampling locations of the susceptibility test observations in panel A.

continent and analysed each data subset independently by fitting separate models (Fig 1 and "Methods"). In West Africa, predicted mean prevalence of resistance to all pyrethroids increased dramatically over the period 2005–2017 (Figs 2 and 3 and S1 Fig, S2 Fig and S3 Fig). Predicted mean proportional mortality to deltamethrin was below 0.9 (the WHO threshold for confirmed resistance) across 15% (95% credible interval [CI] 13%–17%) of the west region in 2005, and across 98% (CI 96.6%–98.7%) of the region in 2017 (Fig 3). These changes in resistance were spatially heterogeneous (Fig 2). Increases in resistance to deltamethrin over the period—in terms of the reductions in the predicted mean proportional mortality—were greatest in northern Liberia (Fig 2D, line A); central Cote d'Ivoire (Fig 2D, line B); the area surrounding the border between Burkina Faso, Cote d'Ivoire, and Ghana (Fig 2D, line C); southern Ghana (Fig 2D, line D); and northern Gabon (Fig 2D, line E). In these regions, resistance to deltamethrin in 2017 was particularly high (with a mean proportional mortality below 0.3 (CI < 0.4).

In East Africa, the prevalence of pyrethroid resistance also increased over the period 2005–2017, albeit at a lesser rate than that in the west region (Figs 2 and 3). Predicted mean proportional mortality to deltamethrin was below 0.9 across 9% (CI 3%–17%) of the east region in 2005 and across 45% (CI 38%–51%) of the region in 2017 (Fig 3). The greatest increases in pyrethroid resistance over the period occurred in the northern part of the region, in the area from central Ethiopia (Fig 2D, line F) westward across most of South Sudan (Fig 2D, line G), and extending into southern Sudan (Fig 2D, line H) and northern Uganda (Fig 2D, line I). Across most of this area, mean mortality to deltamethrin in 2017 was below 0.5 (CI < 0.75). Resistance to deltamethrin increased to a lesser extent in central and southern Uganda, western Kenya, eastern Ethiopia, and coastal Tanzania, with predicted mean mortalities of between 0.6 and 0.8 in these areas in 2017. In areas farther south, differences in predicted resistance over the time period were relatively slight, with mean mortalities changing by less than 0.15 from 2005 to 2017 within Malawi, Mozambique, Zimbabwe, and those parts of Zambia,

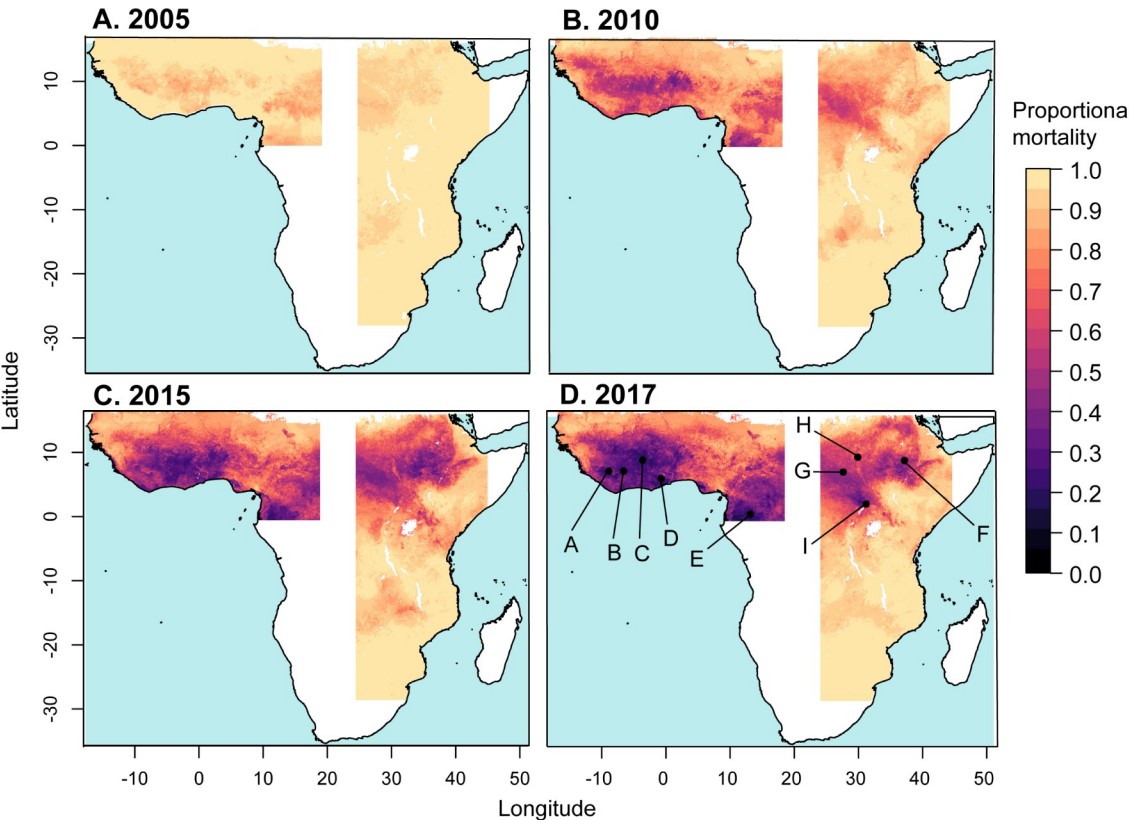

**Fig 2. Predicted mean proportional mortality to deltamethrin across the west and east regions.** (A) 2005, (B) 2010, (C) 2015, and (D) 2017. See 10.6084/m9.figshare.9912623.

Botswana, and South Africa that were included in the model. Similar spatiotemporal trends across the west and east regions occurred in predicted mean resistance to permethrin, λ-cyhalothrin, and α-cypermethrin (S1 Fig, S2 Fig and S3 Fig).

**DDT resistance.** Predicted mean resistance to DDT at the start of the period (in 2005) was more widespread in comparison to pyrethroid resistance and also increased throughout the region from 2005 to 2017 (Figs 4 and 5). In the west region, predicted mean proportional mortality to DDT was below 0.9 across 53% (CI 47%–59%) of the west region in 2005, and across 97% (CI 92.7%–99%) of the region in 2017 (Fig 5). Increases in resistance to DDT over the period were greatest in the area surrounding the border between Liberia and Guinea (Fig 4D, line A), southern Mali (Fig 4D, line B), and central Burkina Faso (Fig 4D, line C). The east region showed a weaker increase in predicted mean resistance to DDT over the period 2005–2017 in comparison to that occurring in the west region. Predicted mean proportional mortality was below 0.9 across 32% (CI 21%–44%) of the east region in 2005, and across 45% (CI 39%–51%) in 2017. Increases in DDT resistance over the period were greatest in central South Sudan (Fig 4D, line D).

## Assessing prediction accuracy

We performed 10-fold out-of-sample validation on the model ensemble (re-running the model 10 times, withholding a different 10% portion of the data each time; see "Methods" and paper by Gelman and colleagues [56]) to assess the accuracy of predicted mean prevalence of resistance. Across all bioassay observations for pyrethroid insecticides, we obtained a root

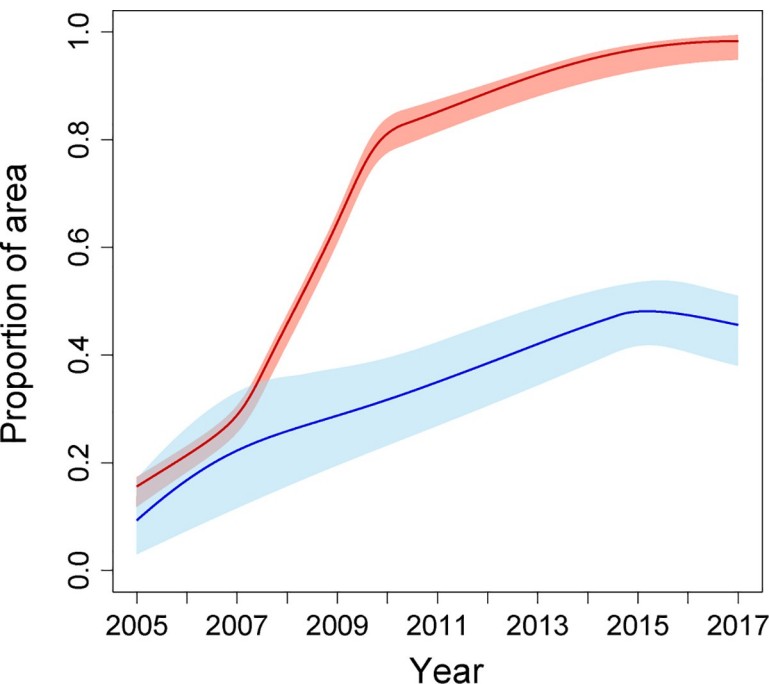

**Fig 3. The proportion of the area with a predicted mean mortality to deltamethrin of less than 0.9, for the west region (red line) and the east region (blue line).** Red and blue shaded areas indicate the 95% CI of the predicted proportion of pixels for the west and east regions, respectively. Numerical values are provided in S2 Data (10.6084/m9. figshare.9912623). CI, credible interval.

mean square error (RMSE) [57] of 0.179 (mean absolute error [MAE] = 0.127; S2 Table) across all the (out-of-sample) predictions of mean proportional mortality for the west and east regions combined (S5 Fig). Across all DDT bioassay observations, the corresponding out-of-sample RMSE was 0.167 (MAE = 0.111; S3 Table). We note that susceptibility tests have a high measurement error, which is reflected by the high values of the data noise parameter estimated by the fitted model [58] (S1 Table) and that the model aims to distinguish spatiotemporal variation in mean resistance that is independent of data noise.

The individual model constituents of our ensemble included 3 machine-learning models: an extreme gradient boosting model (XGB), a random forest model (RF), and a boosted generalized additive model (BGAM). We confirmed that our model ensemble showed improved predictive performance relative to each of these constituent models, as expected (S2 Table and S3 Table). Of the 3 machine-learning models, XGB had the lowest out-of-sample prediction error followed by RF and then BGAM. The fitted mean model weights given by the ensemble model were higher for models with lower out-of-sample prediction error, as expected (S4 Table).

In order to quantify prediction uncertainty across space and time, we produced maps of the 95% CIs of the posterior distributions of predicted mean mortality (Fig 6 and S7 Fig). We again performed 10-fold out-of-sample validation to assess the accuracy of the predicted CIs and found the coverage of the CIs to be accurate when the measurement error associated with the data—estimated by the fitted model [58]—was accounted for (S6 Fig). Prediction error is heterogeneous across space and time, with the 95% CIs of predicted mean mortality being higher in the east compared with the west region and with particularly high CIs in the northwestern part of the east region. This reflects the more sparse distribution of bioassay sampling locations in the east region, particularly in South Sudan and much of southern Sudan (Fig 1).

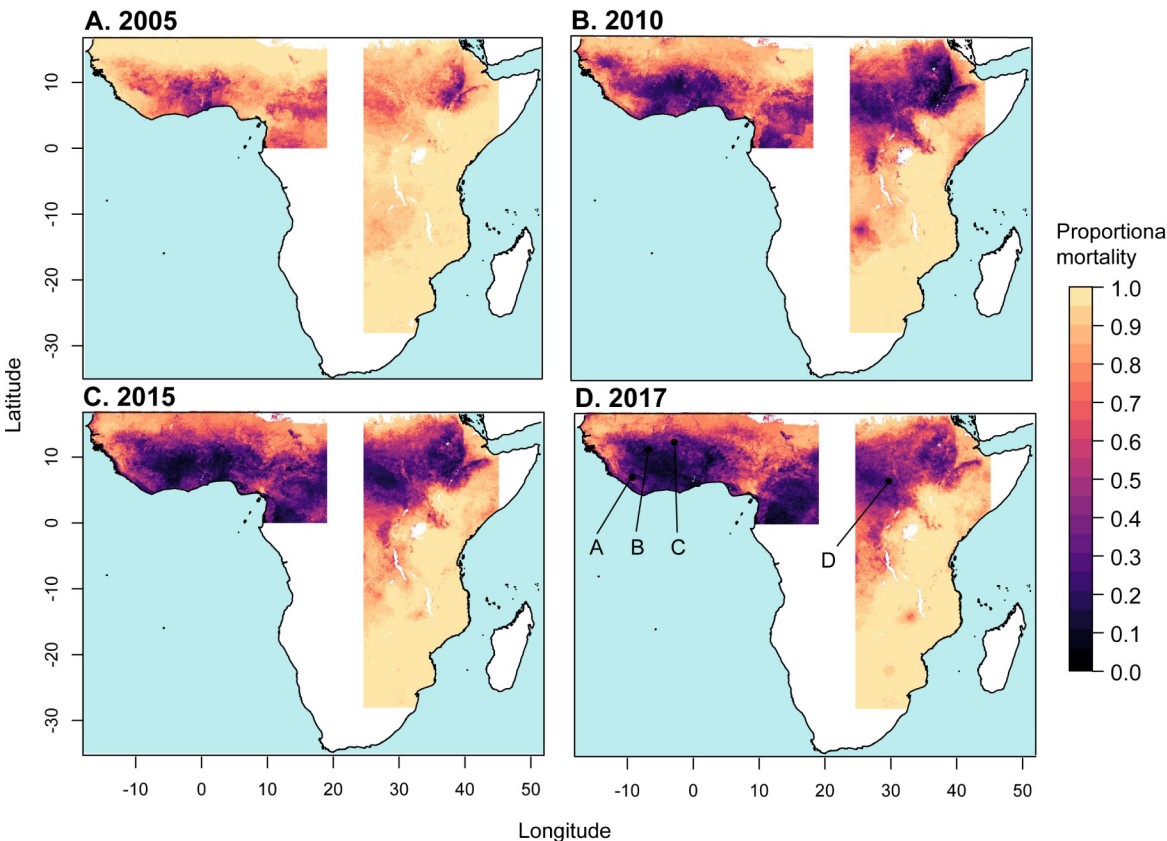

**Fig 4. Predicted mean proportional mortality to DDT across the west and east regions.** (A) 2005, (B) 2010, (C) 2015, and (D) 2017. See 10.6084/m9.figshare.9912623.

**Regional and local resistance trends.** National temporal trends in the predicted prevalence of resistance are qualitatively similar to those occurring in the broader east and west regions, particularly for countries in the west region, which all show a monotonic increase over time in the proportional area with a mean mortality to deltamethrin of <0.9 (Fig 7). Predicted temporal trends are more variable across the countries in the east region, with trends in Kenya, Malawi, Rwanda, and Tanzania suggesting either a plateau or a decline in resistance levels towards the end of the time period (Fig 7). Moreover, predicted temporal trends across point locations show greater variability than regional or national trends, and nonmonotonicity in point trends is common, with declines in resistance occurring following earlier increases (S8 Fig and S9 Fig). Attenuations and declines in resistance may reflect fitness costs of resistance, or they may also arise due to shifts in the composition of the sibling species that make up the *A. gambiae* complex, or immigration of mosquitoes from areas where resistance is lower (see "Discussion").

## Influential predictor variables

Our models used over 100 potential explanatory variables (see the "Methods" section), and our results show which of these variables were most influential to the predictions of mean prevalence of resistance. We obtained measures of variable importance for each of the 3 constituent machine-learning models (XGB, RF, and BGAM). Variable importance measures describe the influence of a variable on model predictions relative to the other predictor variables, but they

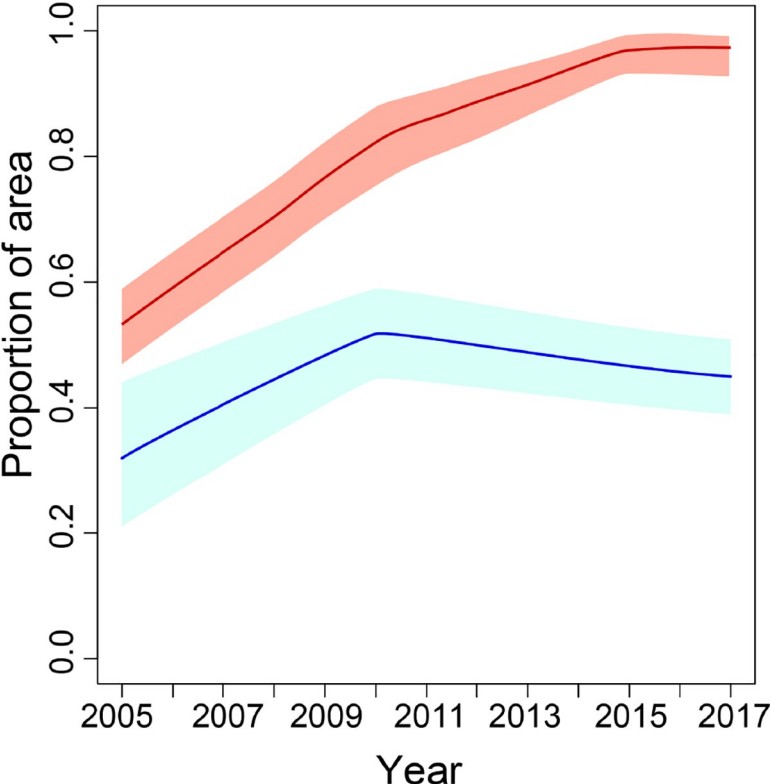

**Fig 5. The proportion of the area with a predicted mean mortality to DDT of less than 0.9, for the west region (red line) and the east region (blue line).** Red and blue shaded areas indicate the 95% CI of the predicted proportion of pixels for the west and east regions, respectively. Numerical values are provided in S2 Data (10.6084/m9. figshare.9912623). CI, credible interval.

can be hard to interpret when predictor variables are correlated (see S1 Text), and they do not identify causal relationships (see "Methods" and "Discussion"). For each model, the importance of each variable is expressed as a fraction of the total importance across all predictor variables. In ranking variable importance, we weighted the importance of each variable given by each model by the model's weight obtained from the Gaussian process meta-model for pyrethroids (S4 Table). This increasingly weights those variables that were more important to models that performed better and thus made a higher relative contribution to the predictions made by the ensemble (Fig 8). Thus, the variable importance values given by XGB and RF are up-weighted relative to those given by BGAM. The original variable importance values produced by each model are given in S5 Table and S6 Table, and a description of each predictor variable is given in S9 Table.

For the west region, variables describing the coverage of ITNs had the highest importance value for each of the 3 models. For XGB and RF, the 3-year lag of ITN coverage had the highest importance value. For BGAM, non-lagged ITN coverage had the highest importance value, and the 3-year lag of ITN coverage had the second highest importance value (Fig 8 and S5 Table). Outside the top two, variables describing climate processes and those describing the area of harvested crops are highly ranked (within the top 20 most important variables) for all 3 models (Fig 8 and S5 Table). For the east region, variables describing ITN coverage and rainfall were ranked in the top 10 most important variables for all 3 models (Fig 8 and S6 Table). More broadly, variables describing climate processes were highly ranked by all 3 models. Our ability

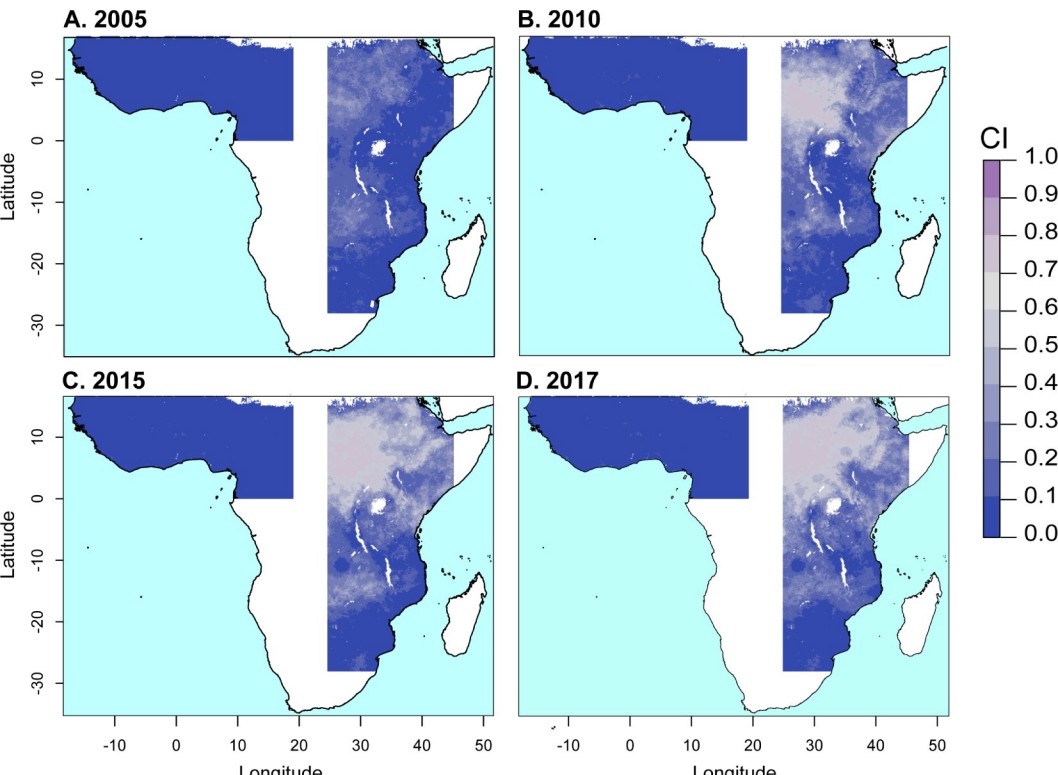

**Fig 6. The prediction error (95% CI) associated with predicted mean mortality to deltamethrin.** See 10.6084/m9. figshare.9912623. CI, credible interval.

to quantitatively compare differences in importance across our set of predictor variables is, however, inhibited by differences in the definition of variable importance used in the different machine-learning approaches that we have employed (see "Methods").

## Discussion

Here, we have quantified spatial and temporal trends in insecticide resistance in the *A. gambiae* species complex in East and West Africa, showing marked increases in the prevalence of resistance to pyrethroids and DDT in recent years, as well as geographic expansion. These results highlight the urgency of identifying and implementing effective resistance management strategies. Our predictive maps of mean prevalence of resistance are available to visualise alongside the latest susceptibility test data on the insecticide resistance mapper website (http://www.irmapper.com) and can guide decisions about resistance management at regional and local levels. In making recommendations, our results will need to be considered in combination with (i) data from resistance monitoring of field samples, including other malaria vector species such as *A. funestus*; (ii) data on the presence of underlying mechanisms of resistance; and (iii) analyses of the expected impacts of resistance management strategies on malaria prevalence [9,59]. Decision-making frameworks also need to explicitly incorporate predictive uncertainty, which is facilitated by our out-of-sample validation results and our mapped Bayesian CIs. Our predictions are not a substitute for ongoing resistance monitoring requirements but highlight areas with particularly high levels of predictive uncertainty, such as parts of South Sudan, southern Sudan, and the Democratic Republic of Congo (Fig 6D). In these

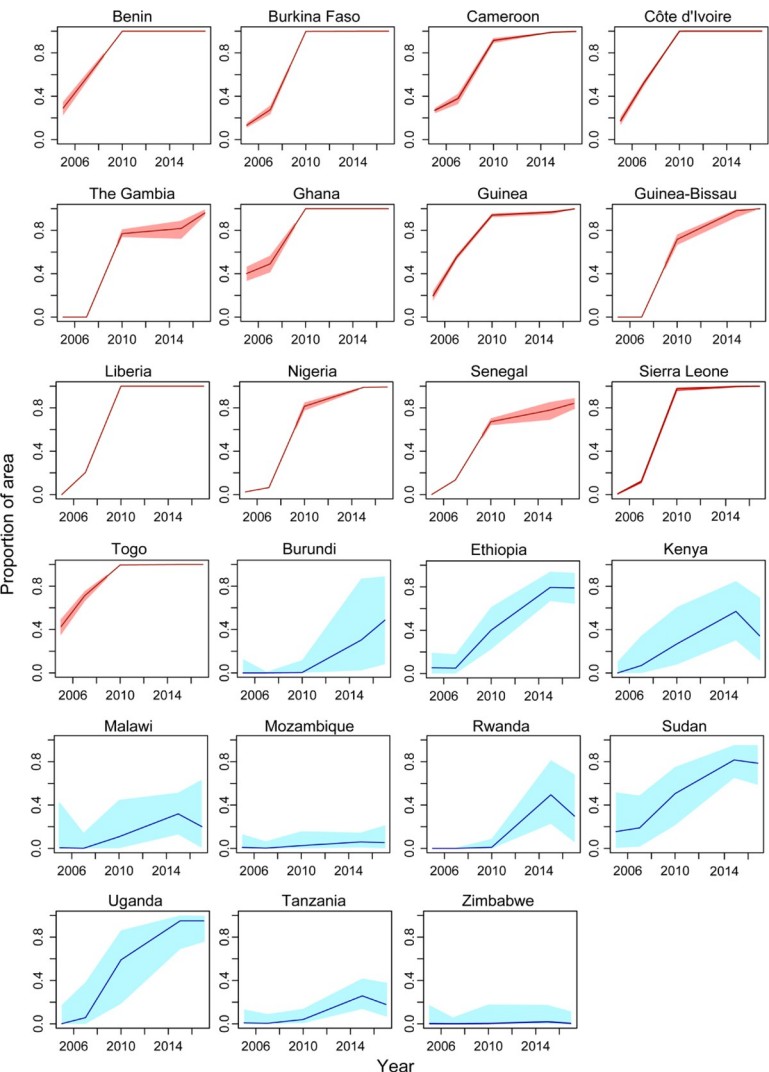

**Fig 7. The proportion of the area with a predicted mean mortality to deltamethrin of less than 0.9 within each country that is fully contained within the west region (red lines) and the east region (blue lines).** Red and blue shaded areas indicate the 95% CI of the predicted proportion of pixels for the west and east regions, respectively. Numerical values are provided in S3 Data and S4 Data (10.6084/m9.figshare.9912623). CI, credible interval.

areas, field sampling to measure resistance is the only means of informing resistance management decisions.

Our results show substantial variation in resistance trends between East and West Africa, as well as within these two regions. Interestingly, ITN coverage was identified as a relatively influential predictor in our models, which is consistent with other studies that have found significant, but spatially variable, increases in pyrethroid resistance associated with the introduction of ITNs [24]. However, in several areas of the central and southern parts of East Africa, such as west Tanzania, ITN coverage has been relatively high (>50%) from 2012 to 2017 [22], but predicted pyrethroid resistance in 2017 was relatively low (Fig 2D). This may be influenced by the locations where resistance mechanisms first emerged, patterns of subsequent gene flow (including restricted flow across the Rift Valley) [41,60,61], and differences among the sibling species within the *A. gambiae* complex [25,30]. For example, the distribution of *A. arabiensis*

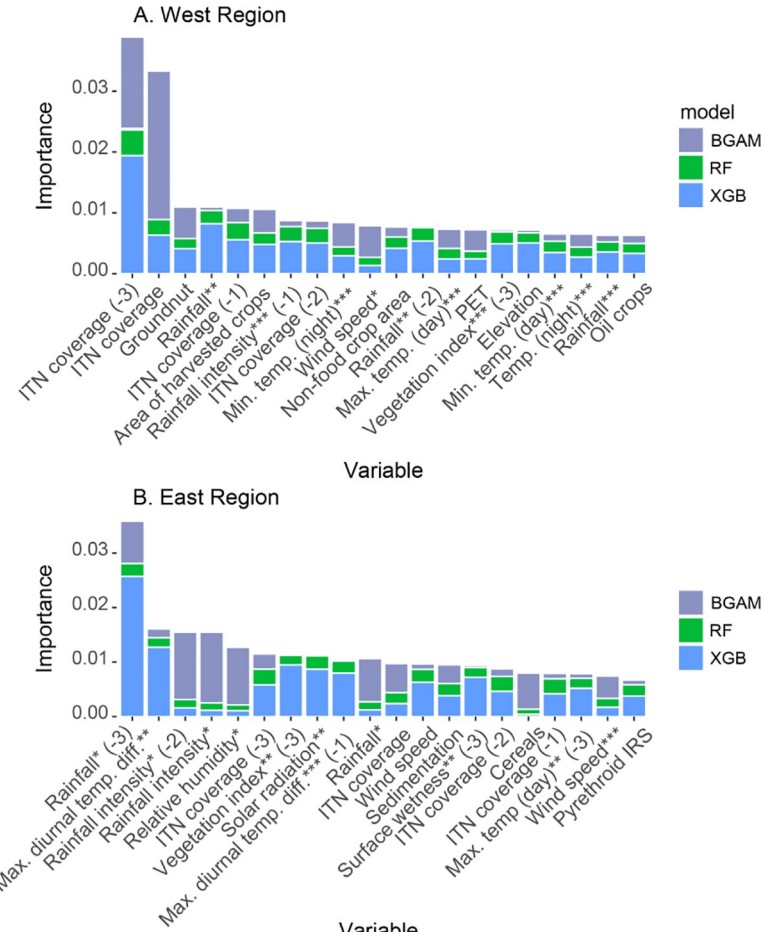

**Fig 8. Weighted variable importance of predictor variables given by the three machine-learning models included in the model ensemble.** (A) West Africa; (B) East Africa. Stacked bars show the relative variable importance given by XGB (blue), RF (green), and BGAM (grey), weighted by the fitted weight for each model given by the Gaussian process meta-model (see text). Variables are ranked by the total height of the stacked bars across the 3 models, and the top 20 variables are shown. The original variable importance values produced by each model are given in S5 Table and S6 Table, and definitions of each predictor variable are given in S9 Table. Variable name suffixes (-1), (-2), and (-3) denote time lags of 1, 2, and 3 years, respectively. One, two, and three asterisks denote the first, second, and third principal component, respectively, for variables available on a monthly time step (see "Methods"). BGAM, boosted generalized additive model; IRS, indoor residual spraying; ITN, insecticide-treated net; PET, potential evapotranspiration; RF, random forest model; XGB, extreme gradient boosting model.

extends further than other species in the complex [33], and this species is known to be more plastic in its feeding behaviour, biting outdoors and feeding on cattle [33]. Therefore, it is possible that selection for resistance in this species lags behind other members of the complex [62–64]. Our predictions of the prevalence of resistance are based on susceptibility tests that often do not identify the sibling species composition of the *A. gambiae* complex sample that was tested. Our analysis only includes test results that are representative of the original sample collected [8, 55], and our predictions cannot directly represent variation in the prevalence of resistance due to variation in the composition of sibling species [33,65]. Routine identification of the composition of sibling species in tested samples—and the provision of species-specific mortality values—would improve the capacity of susceptibility test data to inform prediction of resistance.

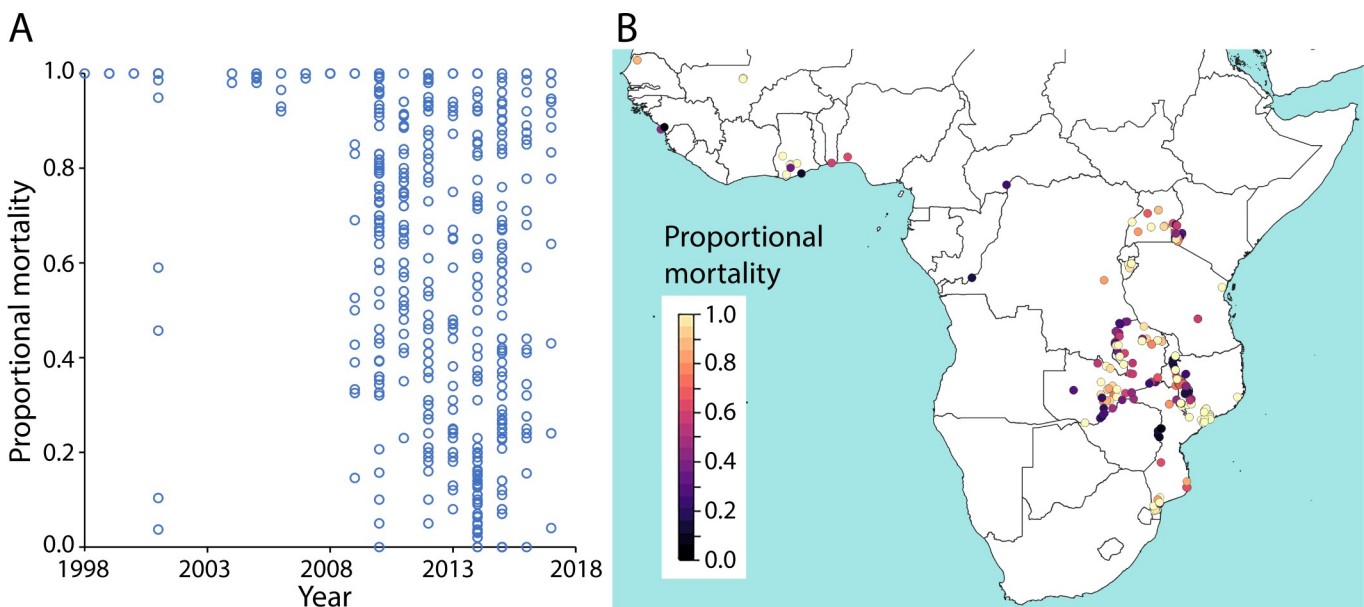

**Fig 9. The sampling distribution of the pyrethroid susceptibility test observations for *A. funestus* mosquito species across space and time.** (A) The proportional mortality to pyrethroids in susceptibility tests performed on *A. funestus* in the years 1998–2017 (*n* = 692). Raw data are available in Moyes et al. 2019 [8]. (B) the locations of the samples in panel A and the recorded proportional mortality.

The coverage of pyrethroid IRS was not among the most influential predictors in our models, but only a small fraction of the areas that we modelled (<5% of the west region and <15% of the east region) received pyrethroid IRS between 2005 and 2017 [4]. Thus, our results do not imply that IRS is not important in driving the selection of resistance. IRS can, however, be a useful tool to prevent the spread of resistance and mitigate its effects because the number of options available for IRS mean chemical classes can be rotated through time, applied in a mosaic in space, or combined for use in the same place and time [9].

It is also important to note that, while our models included over 100 potential predictor variables that may influence selection for resistance, it is unlikely that we have captured the full set of causal variables underlying selection. In particular, data on the quantities of insecticides used in agriculture, and where they were applied, were not available [66]. Such information would better inform predictive relationships between resistance and agricultural insecticide use. We note that the relationships between insecticide resistance and the predictor variables represented in our models do not prove causality. Each variable interacts with other variables (S10 Fig and S11 Fig) and possibly with variables not included in our analysis. For example, variables describing climate processes were ranked as influential predictors (Fig 8), but these may delineate broad areas where resistance trends are similar as a result of an unmeasured process, such as mosquito population structure or species composition. More extensive data on the presence of resistance mechanisms—including a wider coverage of voltage-gated sodium channel (*Vgsc*) allele frequencies, as well as metabolic resistance markers [67]—in field populations will aid in predicting and interpreting resistance trends. The similarity in predicted spatiotemporal patterns in resistance across the 4 pyrethroids and DDT (e.g., Figs 2 and 4) suggests common underlying resistance mechanisms [55].

In some areas, predicted temporal trends indicate plateaus or declines in the prevalence of resistance following an increase. Interestingly, the areas that showed the strongest interannual declines in resistance over the time period often experienced strong increases in earlier years (S9 Fig). These oscillatory dynamics may be the result of fitness costs associated with the

introduction of novel resistance mutations, but the extent of such costs in field *A. gambiae* populations is poorly understood [68]. Several resistance mechanisms exist in field mosquito populations, and they are complex polygenic processes that vary geographically and are continuing to evolve [68,69]. Declines in the prevalence of resistance may also result from a shift in the composition of sibling species; for example, increases in the proportion of *A. arabiensis* have been observed following the instigation of LLIN interventions that predominantly target *A. gambiae*, which bite humans indoors [70]. A future focus on African regions where resistance levels are still relatively low may be deliver new insights into how spread is initiated and how it can be mitigated.

While our analysis focuses on pyrethroids, insecticides from other classes such as carbamates and organophosphates are being increasingly used in IRS interventions [4]. The number of available susceptibility test results for insecticides from these classes is relatively low [8], and spatiotemporal analyses of resistance would benefit greatly from increasing the frequency and spatial coverage of sampling and testing. Susceptibility test data are also more limited for *A. funestus*, a major malaria vector in Africa that is widespread and among the dominant vector species [71]. While the available susceptibility test data for *A. funestus* are insufficient to support a geospatial analysis of the kind performed here, we note that the data indicate higher levels of pyrethroid resistance than we have seen for *A. gambiae* s.l. in southern parts of East Africa (Fig 9).

In summary, our results provide an Africa-wide perspective on recent trends in pyrethroid and DDT resistance in *A. gambiae* complex malaria vectors, demonstrating increasingly high prevalence of resistance to the main insecticides used in malaria control. The rapid spread of resistance across large parts of sub-Saharan Africa signals an urgent need to quantify the efficacy of different resistance management strategies and to understand the impact of resistance on malaria transmission and control. Relationships between insecticide resistance and malaria prevalence are currently poorly understood, but there is evidence that resistance can reduce the efficacy of standard pyrethroid-treated LLINs [3], which have played a key role in achieving reductions in malaria prevalence in Africa over 2000–2015 [1]. Our maps show marked broad-scale spatial heterogeneity in resistance, motivating the implementation and assessment of a wide range of strategies that target different insecticide resistance and malaria transmission settings, such as next-generation LLINs [3,9] as well as rotating insecticide use across different insecticide classes [9].

## Methods

### Data

**Insecticide resistance bioassay data.** Insecticide resistance bioassay data were obtained from a published database [8], which is an updated version of the data used by Hancock and colleagues [55] that includes samples tested up until the end of 2017. The data record the number of mosquitoes in the sample and the proportional sample mortality resulting from the bioassay, as well as variables describing the mosquitoes tested, the sample collection site, and the bioassay conditions and protocol. We used this information to select a subset of records for inclusion in our study (S1 Text). In summary, we include bioassay results for which standard WHO susceptibility tests or CDC bottle bioassays using any one of the 4 pyrethroid types (deltamethrin, permethrin, λ-cyhalothrin, and α-cypermethrin) or the organochlorine DDT was performed on mosquito samples belonging to the *A. gambiae* species complex. We include results from bioassays conducted over the period 2005–2017. Due to spatial heterogeneity in the sampling distribution, we confine our analysis to samples collected from within 2 separate geographic (west and east) regions of sub-Saharan Africa (see Fig 1 and S1 Text). We excluded

Madagascar from our analysis, as our models of resistance on the mainland may not generalize well to island populations. The final number of proportional mortality observations across all insecticide types was 6,423 across 1,466 locations, with 3,515 and 2,908 observations in the west region and east region, respectively (S7 Table and S8 Table).

**Vgsc allele frequency data.** The *Vgsc* is the target site for both pyrethroids and DDT, and mutations in this channel confer resistance. Our analysis used data on the frequency of *Vgsc* mutations in mosquito samples belonging to the *A. gambiae* species complex collected from within the west and east regions over the period 2005–2017 [8,55]. These data record the combined frequency of the single point mutations L1014F and L1014S with respect to the wild-type allele L1014L and comprise 316 observations (215 observations for the west region and 101 observations for the east regions; S7 Table). As described subsequently, we incorporated these data into machine-learning models in order to inform prediction of phenotypic resistance to DDT and pyrethroids by exploiting the positive association between the frequency of *Vgsc* mutations and the prevalence of these resistance phenotypes [55].

**Potential predictor variables.** Our set of predictors includes 111 variables describing environmental characteristics that could potentially be related to the development and spread of insecticide resistance in populations of *A. gambiae* complex mosquito species (described in S9 Table and S1 Text). These variables describe the coverage of insecticide-based vector control interventions, agricultural land use [72,73], and the environmental fate of agricultural insecticides [66], other types of land use [72,74–76], climate [72,77,78], and relative species abundance. Our vector control intervention data include a variable estimating the yearly coverage of ITNs [22,79] and a variable estimating the coverage of IRS with either pyrethroids or DDT year [4]. Relative species abundance is represented by a variable estimating the abundance of *A. arabiensis* relative to the abundance of *A. gambiae* and *A. coluzzii* [65]. For all variables, we obtained spatially explicit data on a grid with a 2.5 arc-minute resolution (which is approximately 5 km at the equator) covering sub-Saharan Africa. For variables for which temporal data were available on an annual resolution, we included time-lagged representations with lags of 0, 1, 2, and 3 years.

## Gaussian process stacked generalization ensemble modelling approach

Stacked generalization is a method of combining an ensemble of models to produce a meta-model, with the aim of achieving better predictive performance than the individual model constituents [80,81]. Here, we adopt a stacking design whereby a set of individual models that make up the first layer, referred to as the "level 0 models," feed into a single meta-model on the second layer, referred to as the "level 1 model." We use the Gaussian process stacked generalization approach developed by Bhatt and colleagues [53], which uses Gaussian process regression as the level 1 model that combines weighted out-of-sample predictions from a set of multiple level 0 models derived from machine-learning methods. The approach exploits the known strengths of these different methodologies, using machine-learning methods to extract as much predictive power from the covariates as possible, and then allowing the Gaussian process to model the spatiotemporal error covariance structure, aiming to further improve prediction. Bhatt and colleagues [53] showed that, under the (restrictive) assumption that the true function is a part of the model's function space, the use of the Gaussian process model of residual variation improves prediction accuracy compared with a standard constrained weighted mean across the ensemble predictions.

**Machine-learning models.** Our set of level 0 models consists of 3 different types of machine-learning model that predict insecticide resistance, using our bioassay mortality observations as the label and our suite of intervention, agriculture, and environmental covariates as

features. The machine-learning approaches employed include an XGB model (implemented using the R package xgboost), an RF model (implemented using the R package randomForest), and a BGAM model (implemented using the R package mboost). We chose these methods because of their demonstrated high predictive performance, particularly in previous applications of Gaussian process stacked generalization to spatial processes [53]. The label for the level 0 models was the proportional mortality observations from bioassays conducted using the 4 pyrethroid types (deltamethrin, permethrin, λ-cyhalothrin and α-cypermethrin), the proportional mortality observations for bioassays conducted using DDT, and the observations of the combined frequency of the *Vgsc* mutations L1014F and L1014S. We included in the label our data on the observed combined frequency of *Vgsc* mutations in mosquito samples, because these observations are significantly associated with the prevalence of resistance to DDT and pyrethroids [55] and can therefore inform prediction of these mortality values. Before performing parameter tuning on the level 0 models, we applied 2 data transformations to the label, the empirical logit transformation followed by the inverse hyperbolic sine (IHS) transformation [82].

The features used in the models included the 111 environmental predictor variables together with the 1-, 2-, and 3-year lags for those variables that vary temporally (on a yearly time step). A factor variable grouping the label according to the type of observation was also included as a feature, assigning a different group to bioassay observations depending on type of insecticide used and whether a WHO or CDC susceptibility test was used. This factor variable also assigned the *Vgsc* allele frequency observations to a separate group. Finally, the year in which the bioassay and allele frequency samples were collected was also included as a feature.

For each level 0 model, parameter tuning was performed using *K*-fold out-of-sample validation based on subdividing the data into *K* training and validation subsets (see S1 Text). In applying the XGB method, we used the DART boosting methodology to avoid overfitting [83].

**Model stacking and Gaussian process regression.** Let $g_A(\mathbf{s}_i, t)$ denote the (empirical logit and IHS-transformed) proportional mortality record for a bioassay using insecticide type *A* conducted on a sample collected at geographic coordinates $\mathbf{s}_i$ and sampling time *t*. To implement Gaussian process stacked generalization, we model the transformed observations, denoted $g_A(\mathbf{s}_i, t)$, using a Gaussian process regression formulation:

$$g_A(\mathbf{s}_i, t) = \mathbf{w}_A \mathbf{M}_{s,t}^A + f_A(\mathbf{s}_i, t) + e_A \tag{1}$$

in which $\mathbf{w}_A$ is a constant vector, $\mathbf{M}_{s,t}^A$ is a design matrix, $f_A(\mathbf{s}, t)$ is a Gaussian process modelled by a spatiotemporal Gaussian Markov random field (GMRF) [84], and $e_A$ is Gaussian white noise $N(0, \sigma_A^2)$. We define a Bayesian hierarchical formulation for the model (Eq 1) using a vector of prior probability distributions for the hyperparameters $\theta_A = [\mathbf{w}_A, \psi_A, \sigma_A]$ in which $\psi_A$ are the parameters of $f_A(\mathbf{s}, t)$ (see S1 Text). To fit the model, the elements of the design matrix $\mathbf{M}_{s,t}^A$ are set to the out-of-sample predictions of the level 0 models derived from *K*-fold cross-validation, i.e., $M_{i,p}^A = \tilde{g}_{A,p}(\mathbf{s}_i t)$, in which $\tilde{g}_{A,p}(\mathbf{s}_i t)$ is the prediction of the $i^{ith}$ withheld (transformed) observation $g_A(\mathbf{s}_i, t)$ given by the $p^{th}$ level 0 model. Validation folds were randomly selected from the full data set. Posterior distributions of $\theta_A$ and $f_A(\mathbf{s}, t)$ are then estimated by fitting the model (Eq 1) using the R-INLA package (www.r-inla.org) [85]. The posterior mean of the vector $\mathbf{w}_A$ contains the fitted weights for each model, representing the relative contribution of each model to the predictions made by the model ensemble. Our implementation of Gaussian process regression (Eq 1) constrains each weight to be positive ($w_p \geq 0, \forall p$) [86]. Once the parameter estimation has been performed, the final set of predictions, $\hat{g}_A(\mathbf{s}, t)$, given by the stacked model are obtained by replacing the elements of $\mathbf{M}_{s,t}^A$ with the in-sample predictions of

the l0 models obtained by fitting each of these models to all the data (all the labels and the corresponding sets of features) [53] (S1 Text).

**Posterior validation.** We performed posterior validation of the stacked model using 10-fold out-of-sample cross-validation, whereby the data were divided into 10 subsets (or "test" sets, using random sampling without replacement), and 10 successive model fits were performed, each withholding a different test set. Each test set was withheld from both the level 0 and level 1 models. We used these out-of-sample predictions to assess the accuracy of the predicted means of the observations as well as their predicted CIs (S1 Text). We also assessed the suitability of our assumed data-generating process using probability integral transform (PIT) histograms on out-of-sample data (S1 Text).

**Predictor variable importance.** We calculated measures of the importance of each predictor variable for each of the machine-learning models used in our model ensemble. For XGB, we used the gain measure calculated for each variable using the xgboost package [87], which is the fractional total reduction in the training error gained across all of that variable's splits. For RF, we used the permutation importance measure calculated using the randomForest package [88], which is the fractional change in the out-of-bag error when the variable is randomly permuted. In the case of BGAM, we used the mboost package [89] to calculate variable importance as the total reduction in the training error across all boosting iterations in which that variable was chosen as the base learner. For each model, we express the importance of a single variable as a fraction of the total importance across all predictor variables in that model.

## Code availability

R code for implementing the XGB, RF, and BGAM models, as well as the R-INLA models for Gaussian process stacked generalisation, is available on GitHub at 10.5281/zenodo.3751786 [90].

## Supporting information

**S1 Fig. Predicted mean proportional mortality to permethrin across the west and east regions.** (A) 2005, (B) 2010, (C) 2015, and (D) 2017. See 10.6084/m9.figshare.9912623. (TIF)

**S2 Fig. Predicted mean proportional mortality to λ-cyhalothrin across the west and east regions.** (A) 2005, (B) 2010, (C) 2015, and (D) 2017. See 10.6084/m9.figshare.9912623. (TIF)

**S3 Fig. Predicted mean proportional mortality to α-cypermethrin across the west and east regions.** (A) 2005, (B) 2010, (C) 2015, and (D) 2017. See 10.6084/m9.figshare.9912623. (TIF)

**S4 Fig. Histograms of the approximate cross-validated PIT values comparing observations and cumulative predictive densities across all susceptibility test observations for pyrethroids.** Numerical values are provided in S5 Data (10.6084/m9.figshare.9912623). (TIF)

**S5 Fig. Predictions of mean proportional mortality from 10-fold out-of-sample validation performed on the Gaussian process regression meta-model.** The vertical axis shows the corresponding value observed from the bioassay. Values for all data points for all pyrethroid types (deltamethrin, permethrin, λ-cyhalothrin, and α-cypermethrin) for the west region (red markers) and the east region (blue markers) are shown. The RMSE across all data values is 0.179

(RMSE = 0.191 for the data within the west region and RMSE = 0.166 for the data within the east region). Numerical values are provided in S6 Data (10.6084/m9.figshare.9912623). RMSE, root mean square error.
(TIF)

**S6 Fig. The proportion of withheld data points that fell within the predicted CIs, based on 10-fold out-of-sample validation, when accounting for the estimated measurement error (see S1 Text).** Numerical values are provided in S7 Data (10.6084/m9.figshare.9912623). CI, credible interval.
(TIF)

**S7 Fig. The prediction error (95% CI) associated with predicted mean mortality to DDT.** See 10.6084/m9.figshare.9912623. CI, credible interval.
(TIF)

**S8 Fig.** The predicted mean proportional mortality to deltamethrin over time for the point locations in the east (A) and west (B) regions that experienced the greatest overall increase in resistance from 2005 to 2017 (Fig 2; locations A, B, C, D, E, F, G, H, and I). Dashed lines show the 95% CIs of the predicted mean mortality. Numerical values are provided in S8 Data (10.6084/m9.figshare.9912623). CI, credible interval.
(TIF)

**S9 Fig.** The maximum interannual change in the predicted mean mortality to deltamethrin over the time period 2005–2017 at each location within the west and east regions: (A) the maximum interannual decrease, (B) the maximum interannual increase. Interannual increases and decreases in predicted mortality are calculated as the difference in predictions between 2 consecutive years, for all years 2005 to 2017. See 10.6084/m9.figshare.9912623.
(TIF)

**S10 Fig. The Pearson correlation coefficient between each of the 20 variables with the highest weighted variable importance value for the models fitted to the West Africa data set.**
(TIF)

**S11 Fig. The Pearson correlation coefficient between each of the 20 variables with the highest weighted variable importance value for the models fitted to the East Africa data set.**
(TIF)

**S1 Table. Fitted parameters of the Bayesian Gaussian process regression models.** Numbers in brackets are the 95% CIs. CI, credible interval.
(DOCX)

**S2 Table. The RMSE given by 10-fold out-of-sample validation performed on the Gaussian process regression meta-model fitted to the bioassay records for the four pyrethroid insecticides (deltamethrin, permethrin, λ-cyhalothrin, and α-cypermethrin) and each of the 3 machine-learning model constituents.** The unit of the transformed RMSE values corresponds to the (empirical logit and IHS-transformed) observations to which the models were fitted. IHS, inverse hyperbolic sine; RMSE, root mean square error.
(DOCX)

**S3 Table. The RMSE given by 10-fold out-of-sample validation performed on the Gaussian process regression meta-model fitted to the bioassay records for DDT and each of the 3 machine-learning model constituents.** The unit of the transformed RMSE values corresponds to the (empirical logit and IHS-transformed) observations to which the models were fitted.

IHS, inverse hyperbolic sine; RMSE, root mean square error.
(DOCX)

**S4 Table. The fitted weights for each constituent model included in the Gaussian process regression meta-model.**
(DOCX)

**S5 Table. Variable importance values for predictor variables given by each machine-learning model included in the model ensemble for the west region.** The 30 variables that were most highly ranked by XGB are shown. Definitions of each predictor variable are given in S9 Table. Variable name suffixes (-1), (-2) and (-3) denote time lags of 1, 2, and 3 years, respectively. One, two, and three asterisks denote the first, second, and third principal component, respectively, for variables available on a monthly time step. XGB, extreme gradient boosting model.
(DOCX)

**S6 Table. Variable importance values for predictor variables given by each machine-learning model included in the model ensemble for the east region.** The 30 variables that were most highly ranked by XGB are shown. Definitions of each predictor variable are given in S9 Table. Variable name suffixes (-1), (-2) and (-3) denote time lags of 1, 2, and 3 years, respectively. One, two and three asterisks denote the first, second, and third principal component, respectively, for variables available on a monthly time step. XGB, extreme gradient boosting model.
(DOCX)

**S7 Table. Number of bioassay records for each insecticide type and number of *Vgsc* allele frequency observations.**
(DOCX)

**S8 Table. Number of bioassay records for each year for each insecticide class.**
(DOCX)

**S9 Table. Descriptions of each potential explanatory variable used in the ensemble model.** If the data layer was obtained from an online repository, the URL and date accessed are given. If the data layer has a citation, then this is given.
(DOCX)

**S10 Table. *l0* models and parameters.**
(DOCX)

**S1 Text. Supplementary information about the modelling methodology.**
(PDF)

## Acknowledgments

The authors are extremely grateful to the many people who contributed unpublished data sets and to the authors who provided additional information linked to their published works.

## Author Contributions

**Conceptualization:** Penelope A. Hancock, Chantal J. M. Hendriks, Janet Hemingway, Michael Coleman, Peter W. Gething, Catherine L. Moyes.

**Data curation:** Chantal J. M. Hendriks, Julie-Anne Tangena, Harry Gibson, Catherine L. Moyes.

**Formal analysis:** Penelope A. Hancock, Ewan Cameron, Samir Bhatt.

**Funding acquisition:** Catherine L. Moyes.

**Investigation:** Penelope A. Hancock, Catherine L. Moyes.

**Methodology:** Penelope A. Hancock, Chantal J. M. Hendriks, Catherine L. Moyes.

**Project administration:** Catherine L. Moyes.

**Resources:** Catherine L. Moyes.

**Supervision:** Peter W. Gething, Catherine L. Moyes.

**Validation:** Penelope A. Hancock.

**Visualization:** Penelope A. Hancock.

**Writing – original draft:** Penelope A. Hancock, Catherine L. Moyes.

**Writing – review & editing:** Penelope A. Hancock, Janet Hemingway, Catherine L. Moyes.

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
