## [Editor Report · Decision Letter 0]

31 Dec 2019

Dear Dr Hancock, 

Thank you for submitting your manuscript entitled "MAPPING TRENDS IN INSECTICIDE RESISTANCE PHENOTYPES IN AFRICAN MALARIA VECTORS" for consideration as a Research Article by PLOS Biology.

Your manuscript has now been evaluated by the PLOS Biology editorial staff as well as by an academic editor with relevant expertise and I am writing to let you know that we would like to send your submission out for external peer review.

Please re-submit your manuscript by Jan 07 2020 11:59PM.

***Please be aware that, due to the voluntary nature of our reviewers and academic editors, manuscripts may be subject to delays due to their limited availability during the holiday season. Please also note that the journal office will be closed entirely 21st- 29th December inclusive, and 1st January 2020. Thank you for your patience.***

Kind regards,

Lauren A Richardson, Ph.D

Senior Editor

PLOS Biology

---

## [Decision Letter · Decision Letter 1]

5 Feb 2020

Dear Dr Hancock,

Thank you very much for submitting your manuscript "Mapping Trends In Insecticide Resistance Phenotypes In African Malaria Vectors" for consideration as a Research Article at PLOS Biology. Your manuscript has been evaluated by the PLOS Biology editors, an Academic Editor with relevant expertise, and by several independent reviewers.

As you will read, the reviewers appreciated many aspects of your work and the importance of these maps. They also raise a number of points that will need to be addressed in a revision. Of particular note, we like the suggestion of Rev #2 to add the data of An.funestus resistance, if it is available. Rev #3 also raises a number of excellent points. We agree that these type of additional analyses would improve the manuscript, but do not consider them to be absolutely essential for a revision. We would like you to consider them and address them as best you can, providing clear explanation for those analyses you do not choose to include in the revision. 

In light of the reviews (below), we will not be able to accept the current version of the manuscript, but we would welcome re-submission of a much-revised version that takes into account the reviewers' comments. We cannot make any decision about publication until we have seen the revised manuscript and your response to the reviewers' comments. Your revised manuscript is also likely to be sent for further evaluation by the reviewers.

We expect to receive your revised manuscript within 2 months. 

**IMPORTANT - SUBMITTING YOUR REVISION**

*Re-submission Checklist*

*Published Peer Review*

*PLOS Data Policy*

*Blot and Gel Data Policy*

Sincerely,

Lauren A Richardson, Ph.D

Senior Editor

PLOS Biology

REVIEWS:

Reviewer #1: 

I really enjoyed reading the manuscript MAPPING TRENDS IN INSECTICIDE RESISTANCE PHENOTYPES IN AFRICAN MALARIA VECTORS from Hancock and colleagues. It is well written (although I would spell out all the acronyms, since some reader may not be familiar with this topic) and the methods scientifically sound. The algorithm development and properties are coherent with the aim of the project, i.e. maximize the predictive capacity of the model (an ensemble). I congratulate the authors for the effort and for providing detailed information about the model behavior and predictions. 

The paper is in line with PLOS Biology scope and aims and I suggest acceptance after the authors revise it in accordance to my corrections/questions.

Raised points:

Abstract.

Is it possible to quantify the "alarming increase", e.g. "since 2005 we have shown a X fold increase in malaria resistance across…"

Introduction

Line 74 should be "after the introduction of these interventions".

The passage from the paragraph in lines 63-83 to the paragraph in lines 85-98 is a bit abrupt. Is it possible to add a paragraph on how this problem (mapping insecticide resistance) has been dealt before? Also how current models fails in achieving what you have achieved.

Results

If I was a general reader, I would found very difficult the interpretation of the RMSE for the comparison of the 4 models. First of all, in table S2 and S3 keep consistency with names (GAMB or BGAM). In terms of values it is not clear to me why you use different units between the main text and the supplementary. If necessary, please add a sentence in the main text.

If I compare figure 1 and 5, it seems to me that lines F,G,H,I and relative areas (figure 1) may not conclusive since uncertainty is close to 1 or very high (Figure 5). Please comment in the discussions or provide a bit more information. In addition, looking at table S7 and S8 I speculate that uncertainty increases although samples are increasing too. Is this the effect of interventions that by fragmenting the "continuous" resistance, reduces the capacity of the spatio-temporal dependence to explain part of the variation?

I found very interesting the importance of rainfall in your models, especially for East Africa which was affected by two important droughts during your study period. Please can you add a comment on this?

Methods

Variable importance. Instead of using different functions (and packages) to evaluate variable importance for each model, it would not be easier to just remove a variable at time and look at the RMSE?

----------------

Reviewer #2: 

Review PBIOLOGY-D-19-03626R1

The paper describes the spatial and temporal trend of insecticide resistance phenotypes in West and East African region using a large database (Moyes CL et al, Scientific dada, 2019) available in open access. Report of insecticide resistance increase in Africa is not new and several others authors have reported this trend in Africa. What is novel in the paper is the visualisation of this increase in resistance over time and space and differences between the West and East block of countries. The paper also explores and ranks factors that predicts the prevalence of insecticide resistance. This is a clear, concise and a well written manuscript of great importance. 

I have however some major comments

* The results for East Africa is not as convincing than the West Africa one, as the authors omit An.funestus resistance that has a major roles in the Southern part of the continent but is also in the past few years increasingly present in East Africa countries (Burundi, Tanzania, Uganda, Kenya). The An.funestus resistance data are also available and applying the model would provide important information that are missing in the actual paper especially as the models aimed to assist decision making in resistance management. The frequency of insecticide resistance in East Africa is not as widespread than in West Africa and resistance management in this area might be the most successful. 

* The contribution of public health vs agriculture insecticides to selection of resistance is essential to device appropriate resistance management. One of the main limitation concerning the predictors is the lack of information on the insecticide/pesticide use for agriculture. The authors used different variables as a proxy, land use of crops and other land cover, fate of insecticides in the environment, etc. These variables correspond to more than half of the variables included in the model. Variables that cover the widest area modelled (ITN, Rainfall, temperature etc..) seems to be the highest contributors. I am therefore wondering if the apparent lower contribution of agriculture pesticide in An. gambiae insecticide resistance is due to the number of variables that have been included. Could the land use variables representing crops using pesticides the most, be pulled together as a single variable in the model? While I understand the difficulty to get the data on agriculture pesticide and find appropriate proxies, the authors should discuss, more in details, some of their findings and differences observed between West and East Africa. The model seems to suggest that agriculture pesticide use is a more important contributor in West than East Africa. Please comments.

* The authors have discussed the difference in insecticide resistance between An.gambiae and An.arabiensis. I have 4 points, I wanted to raise on this.

o The prediction model gives a good idea of the trends in West Africa as well as predicting factors. The resistance trend in the vectors in West Africa (gamiae/coluzzi) is more homogenous than in the other sibling species in East Africa Arabiensis/gambiae. Could it explain the wider credible interval in the prediction they found in East compare to West Africa?

o The authors include a variable on relative abundance of Arabiensis vs gambiae/coluzzi. which do not seem to contribute to the overall model in East Africa. Please would you be able to explain.

o There are reports showing a shift toward An.arabiensis that seems to be associated with IRS and ITN. Would this and lower insecticide resistance prevalence in An.arabiensis explained why ITN coverage is not as strongly predictive for resistance in East Africa compared to West Africa? 

o Contribution of Rainfall variables, please could you explain why do you think there is a difference in importance between West and East Africa. Could rainfall be a proxy for increase in An.gambiae abundance and therefore insecticide resistance? Or?

* Overall, I would suggest the authors to expand on their discussions especially on the contribution factors and possible explanation of differences between West and East Africa. 

Minor comments,

* Why was the variable "Proportional abundance of An. arabiensis to An.coluzzii/gambiae" considered as static? The authors indicated this variable was based on collection from1985 -2015. Would the variable not be able to predict better the resistance if it was included as a yearly or bi-annual variable?

* The authors mentioned that insecticide use for agriculture was not available therefore different crops and livestock production were used. How were these 30 crops selected? Is it based on known insecticide use? 

* The authors indicate that "Our predictive maps of mean prevalence of resistance are available to visualise alongside the latest susceptibility test data on the IR mapper website (http://www.irmapper.com), and can guide decisions about resistance management at regional and local levels. In making recommendations, our results will need to be considered in combination with (i) data from resistance monitoring of field samples, including other malaria vector species such as An. funestus; (ii) data on the presence of underlying mechanisms of resistance, and (iii) analyses of the expected impacts of resistance management strategies on malaria prevalence". 

Would the maps and predictors be available for each country? Otherwise I am not convinced that the overall maps and results as presented in the paper would be enough to help decision making at national level. 

----------------

Reviewer #3: Tovi Lehmann, signed review

Mapping Trends in Insecticide resistance Phenotypes in Africa malaria vectors

By: Hancock et al.

Summary:

This is a well written, interesting paper that I enjoyed reading. It is based an analytic modeling to describe the change of resistance to pyrethroids in the African malaria mosquito, A. gambiae s.l.. Using ~6500 test of mosquito resistance assays performed between 2005 and 2017 and incorporating >100 explanatory variables including application of the pesticides, cross-resistance between insecticide types (including DDT), climate, hydrology, vegetation cover, mosquito species composition, etc., the model generated predictive dynamic (time stamped) maps that describe the level of resistance across large parts of West and East Sub-Saharan Africa. Consistent with other studies, the results confirmed that resistance to Pyrethroids dramaticall increased (from 15% to 98%) over that time in W Africa, but from 9 to 45% in E Africa. Resistance to DDT also increased following similar pattern but to a lesser extent, given the higher starting rates of resistance to DDT in the area. The authors also assess the factors that explain the model and thus likely to affect the evolution of resistance. This analysis highlights the complexity of the factors and difficulty in narrowing the number of key factors down. Among the many effects, coverage of ITNs was highest in W Africa although it was modestly important (and ranked 8th) in E Africa. Yet, it appears that the success of the model depends on small effects of >100 factors or nearly so. The main strength of the study is the production of regional maps that can guide decisions about the management of resistance in areas where few or no tests of resistance were done. However, the authors caution that their predictions are no replacement for such tests and mostly call attention to low confidence indicator for areas where more tests are needed. 

I applaud the authors for a valuable advance in the management of insecticide resistance and their application of state-of-the-art computing to generate these impressive maps. The careful interpretation of the results is also a strength of this paper. Yet, I feel that this paper better fits a journal with more specialized readership in a narrower field of malaria epidemiology than PlosBiology. I note below that if the authors extend their analyses to address additional new questions, then their work could well fit with PlosBiology's broad readership. I hope my comments below will help the authors decide about the best course for their paper.

General Comments:

1. Given the exceptional analytic skills and unique data, I wonder if the authors would focus on some additional questions (using regions/time sections that are suitable to address them as applicable) that will "elevate the interest" in their work. For example (a) Are true "reversals of resistance" evident? How to discern true reversals? If observed, how common are they? What factors may account for them? Are they more common during early evolution phase (when mutations are presumably more costly) and how they can be exploited in management of resistance?

b) are there islands of susceptibility and resistance that are stable. If so where, how long, and what could account for them (see above). 

c) is there a way to infer the "lower rate of spatial spread of resistance" (assuming arriving from nearest area) over distance and possibly certain barriers?

d) how unstable are resistance profiles at a location, so if we have several resistance tests in a region that had indicated 51-67% 2,3,4, or up to 7 years ago, how important is to do that test this year if there is no dramatic change in insecticide use and if there was. How such answers change based on availability of a complete time series 10, 50, 100, or 500 km away. So, can we refine the guidance when and where we need new tests. 

2. The term "spread of resistance" is complex and may better replaced here or be explained in the outset. Intuitively, it conveys the notion of (1) spread from point A to B, which is not the case here. The authors include (2) the rise in frequency of resistance in the same region, and (3) the spread from unknown multiple sources so that no point A to B can be given. A clarification would be helpful.

3. Maps are very good to convey "big picture" of changes in resistance level over time but seem a little too perfect and convey a false notion that all is known. I cannot tell which parts of a map are close to "observed", which show gaps between observed and predicted (how big the gap), and which are "predicted" with no observed to compare with. 

These are a few examples for questions that once being addressed may provide new insights that will merit publication in a broad audience journal such as PlosBiology.

Specific comments 

Introduction 

- why the range of data used/selected was 2005-17? Given that there is much widespread resistance in 2005, it would be interesting to start earlier, even if it would address smaller areas. 

-. It is mentioned that pyrethroids have been extensively used and given the analysis by different types of pyrethroids, it will help knowing about the difference in sensitivity for each type, mechanism of resistance to each type, and historical use of these types in different areas. Some of this info can form a basis for expectations to judge the maps against. 

- The introduction led me to assume that there were no previous papers on the origin and spread of mechanisms of resistance which were depicted by maps. If this is incorrect, it'd be helpful to refer to their maps and underlying basis. 

Results

I believe if date-stamp is 2005, data were taken only during this year (not from a range, e.g., 2004-2006). Correct?

L 198 -214. - 

- I assume 'out-of-sample' means removing an observation and using the rest of the data to predict it. Is 10-fold means that it was repeated 10 times across all observation, one at a time or 90% of the data were used? It needs to be explained.

- I believe that RMSE should be given side by side with MAE, which is more directly applicable to the data and is less susceptible to outliers.

- I think this stat is more meaningful conditioned on distance (spatial and temporal if there were multiple observations in the time interval and only 1 was removed) from nearest data points. Further, I'd report the 'adjusted' value to reflect the mean and median distance between observed and predicted that the map shows.

- I agree that validation of model prediction is vital and fit in the results, although this section is written too technically, and most readers won't be able to interpret it. I suggest keeping the section but transfer parts to the methods and in their place, provide biological relevant predictions capturing a couple of the better and worse predictions pertaining to relevant space and time points.

---

## [Decision Letter · Decision Letter 2]

31 Mar 2020

Dear Dr Hancock,

Thank you for submitting your revised Research Article entitled "MAPPING TRENDS IN INSECTICIDE RESISTANCE PHENOTYPES IN AFRICAN MALARIA VECTORS" for publication in PLOS Biology. I have now obtained advice from two of the original reviewers and have discussed their comments with the Academic Editor. 

Based on the reviews, we will probably accept this manuscript for publication, assuming that you will modify the manuscript to address the remaining points raised by reviewer #3. Please also make sure to address my Data Policy-related requests noted at the end of this email.

We expect to receive your revised manuscript within two weeks. Your revisions should address the specific points made by each reviewer. In addition to the remaining revisions and before we will be able to formally accept your manuscript and consider it "in press", we also need to ensure that your article conforms to our guidelines. A member of our team will be in touch shortly with a set of requests. As we can't proceed until these requirements are met, your swift response will help prevent delays to publication.

*Copyediting*

*Published Peer Review History*

*Early Version*

*Submitting Your Revision*

Sincerely,

Roli Roberts

Senior Editor

PLOS Biology

DATA POLICY:

Many thanks for depositing your data in Figshare; this is much appreciated. However, my understanding is that this largely comprises the "raw" geographical raster data; we also ask that all the numerical values summarized graphically in the figures and results of your paper be made available in one of the following forms:

Regardless of the method selected, please ensure that you provide the individual numerical values that underlie the summary data displayed in the following figure panels as they are essential for readers to assess your analysis and to reproduce it: Figs 1A, 3, 5, 7, 8, 9, S4, S5, S6, S8. NOTE: the numerical data provided should include all replicates AND the way in which the plotted mean and errors were derived (it should not present only the mean/average values).

Please also ensure that figure legends in your manuscript include information on where the underlying data can be found (i.e. the Figshare deposition and the bioRxiv DOI), and ensure your supplemental data file/s has a legend.

REVIEWERS' COMMENTS:

Reviewer #2:

The authors have addressed all my comments and answered my questions adequately. I am glad they included An.funestus results even if the number of data point were much lower than An.gambiae.

Reviewer #3:

[Identifies himself as Tovi Lehmann]

Overall the revisions are thorough and satisfactorily address all the key points. The sophisticated methodology is explained in less-technical terms and the new graphs help provide a more complete picture. I enjoyed reading the paper and feel it will resonate with much interest. So my recommendation is "Accept".

I understand if the authors prefer to expand on that elsewhere, but I would like to hear their comments (2-3 sentences) on the relationships between insecticide resistance and corresponding malaria decline, or lack of decline over that 13-15 years time frame. Their analysis may provide a very valuable if not a decisive answer to this important yet incompletely addressed question. 

Minor points (up to authors to decide if they would like to address):

- L270-2. 'Attenuations and declines in resistance may reflect fitness costs of resistance, or they may also arise due to shifts in the composition of the sibling species that make up the An. gambiae complex (see the Discussion).' 

I'd consider adding a the contribution of migration from areas with low resistance. 

L 470 'in populations of Gambiae complex mosquito species' , consider revising this fragment. At least, change to ..An. gambiae complex...

---

## [Editor Report · Decision Letter 3]

11 May 2020

Dear Dr Hancock,

On behalf of my colleagues and the Academic Editor, Andrew Fraser Read, I am pleased to inform you that we will be delighted to publish your Research Article in PLOS Biology. 

Early Version

PRESS 

Kind regards,

Alice Musson

Publishing Editor, 

PLOS Biology

on behalf of

Roland Roberts,

Senior Editor

PLOS Biology